# Structure of avian influenza hemagglutinin in complex with a small molecule entry inhibitor

Aleksandar Antanasijevic[1], Matthew A Durst[1], Han Cheng[2], Irina N Gaisina[3], Jasmine T Perez[4], Balaji Manicassamy[4], Lijun Rong[2], Arnon Lavie[1], Michael Caffrey[1]

**HA plays a critical role in influenza infection and, thus HA is a potential target for antivirals. Recently, our laboratories have described a novel fusion inhibitor, termed CBS1117, with $EC_{50}$ ~3 $\mu$M against group 1 HA. In this work, we characterize the binding properties of CBS1117 to avian H5 HA by x-ray crystallography, NMR, and mutagenesis. The x-ray structure of the complex shows that the compound binds near the HA fusion peptide, a region that plays a critical role in HA-mediated fusion. NMR studies demonstrate binding of CBS1117 to H5 HA in solution and show extensive hydrophobic contacts between the compound and HA surface. Mutagenesis studies further support the location of the compound binding site proximal to the HA fusion peptide and identify additional amino acids that are important to compound binding. Together, this work gives new insights into the CBS1117 mechanism of action and can be exploited to further optimize this compound and better understand the group specific activity of small-molecule inhibitors of HA-mediated entry.**

## Introduction

Standard, trivalent influenza vaccines, which are designed to protect against H1N1, H3N2, and influenza B viruses, target the envelope protein HA (Ellebedy & Webby, 2009). However, HA readily mutates, which requires that the vaccine composition be reviewed each year to account for changes in antigenicity and that the effectiveness of the vaccine varies from year to year with average protection rates of 50–60% (Monto, 2010). Consequently, there is much interest in the development of small-molecule antivirals. Current treatments for influenza are limited and include small molecules targeting the M2 channel (Symmetrel and Flumadine [Lagoja & De Clercq, 2008; Yen, 2016]), neuraminidase (NA) (e.g., Tamiflu [Lagoja & De Clercq, 2008; Yen, 2016]) and most recently the cap-dependent endonuclease (Xofluza [Yang, 2019]). In the case of

M2 channel inhibitors, they are rarely given because of wide-spread resistance in circulating strains. Similarly, many circulating strains are resistant to current NA inhibitors. For example, the 2008–2009 H1N1 strain exhibited ~100% resistance against Tamiflu (van der Vries et al, 2010), and there are reports of Tamiflu resistance in some avian H5N1 and H7N9 strains (Skeik & Jabr, 2008; Liu et al, 2013). Furthermore, with respect to the newly developed antiviral targeting, the cap-dependent endonuclease, the emergence of resistance in humans after a single dose of Xofluza is troubling (Yang, 2019). Taken together, the challenges of vaccine design and the limited efficacy of current antivirals underscore the importance of novel influenza treatments.

Infection by influenza requires the viral envelope protein HA, which mediates entry of the virus into the appropriate target cells through a series of orchestrated steps (Wiley & Skehel, 1987; Skehel & Wiley, 2000; Eckert & Kim, 2001; Harrison, 2008). During viral maturation, HA is glycosylated, and it assembles into a homotrimer anchored to the membrane by a transmembrane domain. In addition, each HA protomer is processed by cellular proteases to form HA1 and HA2 subunits, which remain associated through noncovalent interactions and a disulfide bond (Wiley & Skehel, 1987). The processing frees the N terminus of HA2, which allows this region to play its key role in viral entry as the "fusion peptide" (Wiley & Skehel, 1987; Skehel & Wiley, 2000). In the initial step of viral entry, the HA1 subunit binds to sialic acid moieties present on the target cell surface, and the virus is internalized via endocytosis. Subsequently, the pH of the endosome is acidified, which triggers the loop to helix transition in the "stem loop" region of HA2 (Carr et al, 1997), resulting in a large conformational change from the "neutral pH structure" to the "low pH structure" of HA (Wiley & Skehel, 1987; Skehel & Wiley, 2000). It is at this point that the HA2 "fusion peptide" becomes inserted in the endosomal membrane and, after a further refolding event, HA2 mediates fusion of the viral and target membranes, thereby allowing the release of the viral RNA into the cytoplasm. HA plays a critical role in influenza entry and consequently is a potential target for antivirals (Wu et al, 2017; Wu & Wilson, 2018). Recently, our laboratories have described the discovery of a

---

[1]Department of Biochemistry and Molecular Genetics, University of Illinois at Chicago, Chicago, IL, USA   [2]Department of Microbiology and Immunology, University of Illinois at Chicago, Chicago, IL, USA   [3]Chicago BioSolutions, Inc., Chicago, IL, USA   [4]Department of Microbiology and Immunology, University of Iowa, Iowa City, IA, USA

Correspondence: caffrey@uic.edu
Aleksandar Antanasijevic's present address is Department of Integrative, Structural and Computational Biology, The Scripps Research Institute, La Jolla, CA, USA

HA fusion inhibitor compound with a 4-aminopiperidine scaffold from an HTS screen of ~20,000 compounds (Hussein et al, 2020). The best hit, termed CBS1117, exhibited $EC_{50}$ = 3.0 $\mu M$ and low toxicity ($CC_{50}$ > 100 $\mu M$) in the pseudotype virus assay in A549 cells infected with influenza HA from H5N1 (Gaisina et al, 2020; Hussein et al, 2020). In this work, we characterize the binding of CBS1117 to avian H5 HA by x-ray crystallography, NMR, and mutagenesis and discuss new insights into the compound's mechanism of action and group specific activity.

# Results

### X-ray crystallographic structure of the H5 HA in complex with CBS1117

To understand the structural basis for fusion inhibition and give guidance into future efforts to optimize this class of compounds, we determined the crystal structure of CBS1117 bound to H5 HA (A/Vietnam/1203/04 [H5N1]) at 2.20 Å resolution (Table 1). For this analysis, crystals of the trimeric extracellular domain of H5 HA were soaked in a cryosolution containing 5 mM CBS1117, as described in the Materials and Methods section. Analysis of the resulting electron density maps revealed three CBS1117-binding sites at symmetric locations on the H5 HA trimer (Fig 1A). As shown in Fig 1A, CBS1117 binds near the HA fusion peptide (residues 1–20 of HA2, shown in red) with a closest heavy atom approach of <5 Å but relatively distant from the HA stem loop (residues 59–74 of HA2, shown in blue) with a closest heavy atom approach of >17 Å. CBS1117 binds to the HA surface at a site between the HA1 and HA2 subunits of a single protomer (Fig 1B) with the compound forming contacts with HA1 residues H38, Q40, and T325 and HA2 residues W21, I45, and T49 (Figs 2 and S1). Thus, the compound appears to stabilize the prefusion HA conformation by bridging interactions between the HA1 and HA2 subunits with very little structural change in the side chain conformations of the HA binding pocket (RMSD value of 0.26 Å between apo and compound bound HA for all atoms of 32 residues that are in closest proximity to the binding site). Specific types of intermolecular interactions include hydrophobic (residues HA1-H38, HA1-Q40, HA2-W21, and HA2-I45) and polar interactions between the chlorine atoms of CBS1117 and side chains of residues HA1-T325 and HA2-T49. The importance of the polar interaction between the benzyl 2-chlorine and the HA2-T49 side chain hydroxyl is in agreement with the SAR analysis of this series of compounds, which suggested that halogen substitutions to the benzyl ring were essential for compound activity (Gaisina et al, 2020; Hussein et al, 2020). Finally, we note that characterization of the H5 HA-CBS1117 interaction by WaterLOGSY NMR confirmed compound binding to H5 HA (Fig S2A). Moreover, the STD NMR characterization suggested that the aromatic portion of the molecule is closest to the HA-binding surface and that the piperidine and isopropyl groups are farther from the HA-binding surface (Fig S2B), which correlates well with the $^1$H solvent exposure observed in the crystal, for example, the $^1$H methyl and piperidine groups exhibit relatively high solvent exposure (42 and 36 Å$^2$, respectively) and the $^1$H of the aromatic group exhibit relatively lower solvent exposure (25 Å$^2$).

### Site-directed and resistance mutants confirm the CBS1117-binding site

Previously, we have generated a set of point mutations in H5 HA in the region near the site of CBS1117 binding, and characterized the mutational effects on expression, processing, receptor binding, and viral entry (Antanasijevic et al, 2014b). To verify the CBS1117-binding site on H5 HA, we assessed the effects of point mutations to inhibition of viral entry by the compound at concentration slightly above the $EC_{50}$ using a pseudovirus entry assay (Antanasijevic et al, 2014b). Under these conditions, the wild-type exhibits ~35% entry (Fig 3A, c.f. Hussein et al [2020] for a description of the compound's dose-dependent inhibition). Mutants HA2-Q42A and HA2-N53A exhibit little effect on inhibition (Fig 3A), suggesting that these mutations do not disrupt the interaction between CBS1117 and H5 HA. In contrast, mutants HA2-I45A, HA2-T49A, and HA2-V52A exhibit significantly reduced inhibition, suggesting that these mutations disrupt the binding of CBS1117 to H5 HA (Fig 3A). As shown by Fig 3B, the residues with the largest effects on the compound's efficacy are in relatively close proximity and the residues with the smallest effects on the compound's efficacy are relatively distant. Moreover, HA2-I45 and HA2-T49 form important contacts with CBS1117 (Fig 2B) and thus the observed reduced potency of alanine substitutions at these sites is not surprising. To further establish the CBS1117-binding site, we generated escape mutants in an H1 HA bearing influenza strain (H5 and H1 HA exhibit a high degree of structural homology and thus the HA interactions with CBS1117 are expected to be similar, c.f. Fig S3). Specifically, after nine passages of infectious influenza strain PR8 (H1N1) in the presence of the compound with concentrations of compound increasing every two passages, resistant mutations were found at positions HA1-M323I, HA1-T325I, HA2-N104D, and HA2-F110S (H5 HA numbering). These mutations are very close to the binding site (Fig 3C, residue 323 of HA1 is a leucine in H5 HA). In particular, the HA1-T325I–resistance mutant, as well as the HA1-T325A site-directed mutant discussed above, eliminate a polar interaction with CBS1117 (Fig 2A). In summary, the site-directed mutagenesis and resistance experiments are fully consistent with the protein-compound interactions seen in the x-ray structure (Fig 2A and B), suggesting a similar binding mode in H1 HA.

# Discussion

In this work, we have characterized the interaction between the recently discovered entry inhibitor CBS1117 and avian H5 HA by multiple approaches. The crystal structure shows that CBS1117 binds in a cavity near the fusion peptide and forms hydrophobic and polar interactions spanning HA1 and HA2 subunits of a single protomer, which presumably stabilizes the prefusion conformation of HA and thereby inhibits transition to the fusion state in a manner similar to other fusion inhibitors (Russell et al, 2008; Antanasijevic et al, 2013; Basu et al, 2017). HA1-H38 has recently been proposed by our laboratories to be part of a pH trigger through its interaction with HA1-H18 and relatively close proximity to the fusion peptide (Antanasijevic et al, 2020), and thus, the close proximity of the

**Life Science Alliance**

**Table 1.  Data collection and refinement statistics for H5 HA in complex with Compound CBS1117.**

| Structure | H5 HA in complex with CBS1117 |
|---|---|
| PDB codes | 6VMZ |
| Data collection statistics | |
| X-ray source and detector | LS-CAT (ID-G) MAR CCD 300 |
| Wavelength (Å) | 0.979 |
| Temperature (K) | 100 |
| Resolution (Å)[a] | 2.20 (2.32–2.20) |
| Number of reflections | |
| Observed[a] | 545,596 (81,318) |
| Unique | 108,659 (16,617) |
| Completeness (%) | 92.3 (88.7) |
| $R_{meas}$ (%) | 8.8 (68.6) |
| $CC_{1/2}$ (%) | 99.8 (76.2) |
| Average $I/\sigma(I)$[a] | 11.78 (2.29) |
| Space group | $P2_12_12_1$ |
| Unit cell (Å): a, b, c | 72.71, 126.08, 249.64 |
| (°): $\alpha$, $\beta$, $\gamma$ | 90.00, 90.00, 90.00 |
| Refinement statistics | |
| Refinement program | REFMAC5 |
| $R_{work}$ (%) | 20.79 |
| $R_{free}$ (%) | 25.17 |
| Resolution range (Å) | 124.8–2.2 |
| Protein molecules per a.u. | 6 |
| Number of atoms: | |
| Protein (chain A, chain B, chain C, chain D, chain E, and chain F) | (2548, 1412, 2566, 1412, 2556, and 1146) |
| Water molecules | 435 |
| NAG Ligands | 112 |
| Compound CBS1117 | 60 |
| RMSD from ideal: | |
| Bond length (Å) | 0.0087 |
| Bond angles (°) | 1.5056 |
| Average B-factors (Å²) | |
| Protein (chain A, chain B, chain C, chain D, chain E, and chain F) | (46.4, 63.0, 43.20, 68.6, 49.1, and 73.0) |
| Water molecules | 45.2 |
| NAG Ligands | 93.0 |
| Compound CBS1117 | 93.4 |
| Ramachandran plot (%): | |
| Most favored regions | 97 |
| Additionally allowed regions | 3 |
| Outlier regions | 0 |

[a]Parenthesis denote the highest resolution shells.

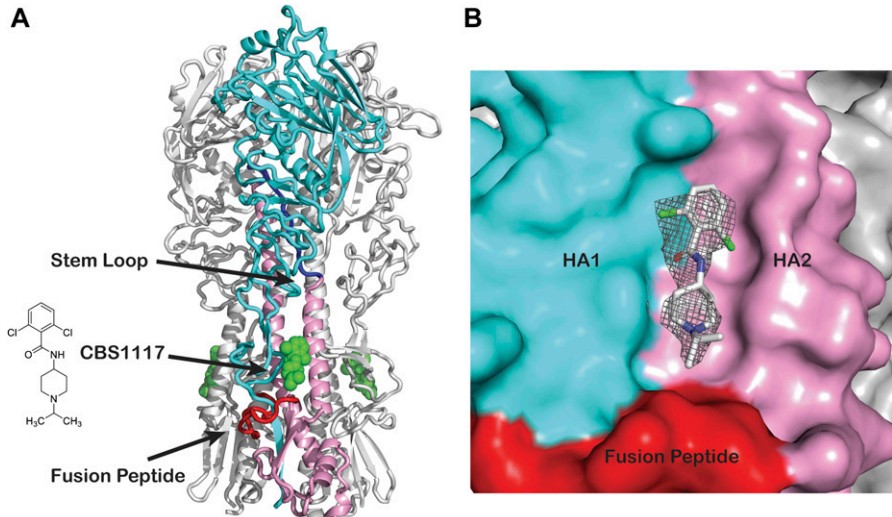

**Figure 1.   Crystal structure of influenza H5 HA in complex with viral entry inhibitor CBS1117.**
**(A)** Ribbon diagram of H5 HA in complex with CBS1117. The compound is shown in green and the HA1 and HA2 subunits of one protomer are colored cyan and pink, respectively. The fusion peptide and stem loop are colored red and blue, respectively. **(B)** H5 HA surface shown with the $2F_o - F_c$ difference (gray mesh, contour level at 1 $\sigma$). The HA1 and HA2 subunits of one protomer are colored cyan and pink, respectively; the fusion peptide is colored red.

CBS1117-binding site to the histidine pair (Fig 2B) may disrupt the triggering event and subsequent membrane fusion. The NMR experiments, performed in solution, confirm binding of CBS1117 to H5 HA and support the compound's relative orientation in complex with H5 HA. As previously discussed by our laboratories, the STD NMR results identify compound regions that exhibit the highest potential for modifications to enhance potency and pharmacological properties (Antanasijevic et al, 2013, 2016). Accordingly, the isopropyl group of CBS1117, which makes the least contact with the protein surface, would seem to be an attractive site for chemical optimization. Finally, site-directed mutagenesis of H5 HA confirms that drug sensitivity depends on residues close to the CBS1117. Moreover, the resistance mutations in H1 HA (another group 1 subtype) selected by propagating the virus in the presence of CBS1117 were at four positions immediately adjacent to the CBS1117-binding site in H5 HA. Thus, the compound appears to bind similarly to HA of both group 1 subtypes.

In recent work, our laboratories have performed chemical optimization of CBS1117 to obtain higher potency against group 1 HA

(Gaisina et al, 2020). The present work allows us to better interpret the structure activity relationship of our compounds. For example, compound 16 (N-(1-(*tert*-butyl)piperidin-4-yl)-2-chloro-4-(trifluoromethyl)benzamide) with bulky and lipophilic *tert*-butyl substituents attached to the piperidine nitrogen and a 2-chloro-4-trifluoromethyl substituent pattern in the aromatic ring appears to be a superior inhibitor with $EC_{50}$ = 0.24 $\mu$M against H5 HA in the pseudovirus entry assay ([Gaisina et al, 2020], compound chemical structure shown in Fig S4). In contrast, Compound 26 ((2,6-dichlorophenyl) (8-isopropyl-1-oxa-4,8-diazaspiro[4.5]decan-4-yl) methanone) with the amide replaced by oxazolidine pharmacophore exhibits $EC_{50}$ > 30 $\mu$M against H5 HA in the pseudovirus entry assay ([Gaisina et al, 2020], compound chemical structure shown in Fig S4). The overlay of compound 16 in the CBS1117 binding site reveals maintenance of the existing interactions and a new potential hydrophobic interaction with the side chain of HA2-V52, which presumably is partially responsible for the 10× enhancement of potency (Fig 4). On the other hand, the overlay of compound 26 in the CBS1117-binding site reveals loss of numerous hydrophobic

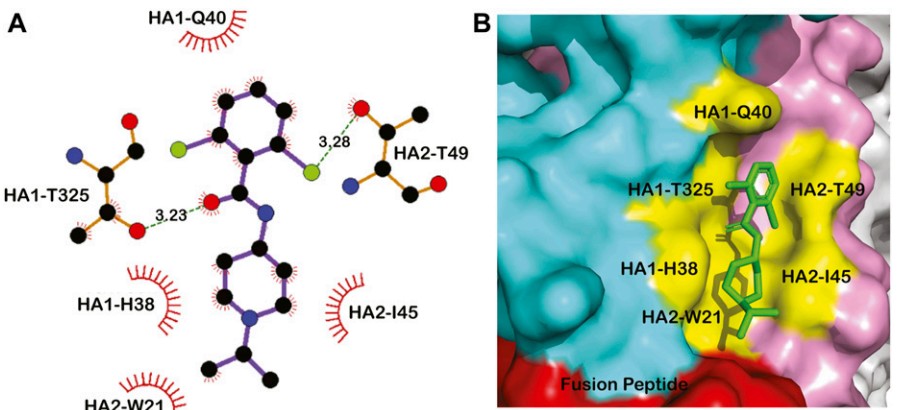

**Figure 2.   Interactions between CBS1117 and influenza H5 HA.**
**(A)** Ligplot analysis of CBS1117 interactions with H5 HA. Polar interactions between the chlorine atoms of CBS1117 and residues HA1-T325 and HA2-T49 are shown as green dashes. **(B)** Surface representation of the interactions between CBS1117 (green) and H5 HA side chains (yellow); the HA1 and HA2 subunits of one protomer are colored cyan and pink, respectively, and the fusion peptide is colored red.

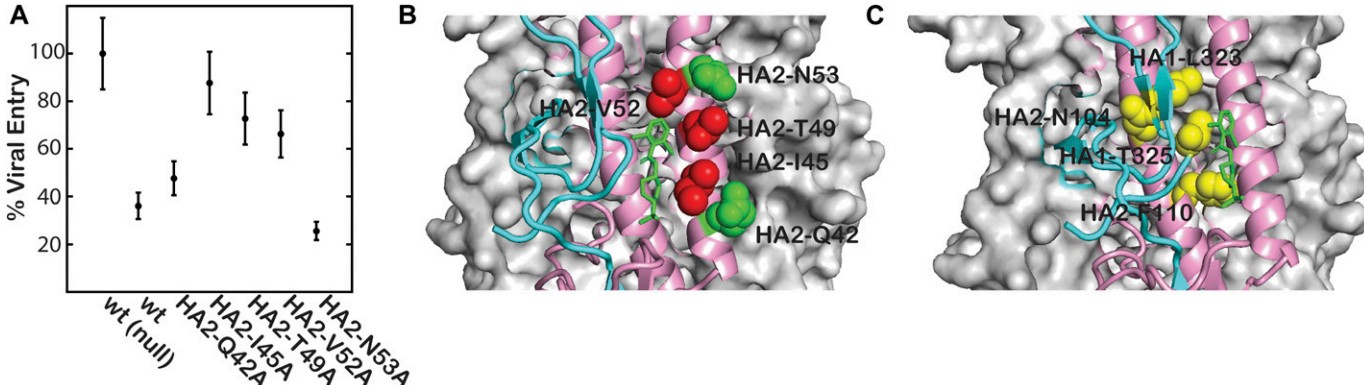

**Figure 3. Mutagenesis studies of the CBS1117-binding site on influenza H5 HA.**
**(A)** Mutational effects on inhibition of H5 HA–mediated viral entry. The relative entry levels are based on the entry of pseudovirus into A549 lung cells using a luciferase-based assay ±CBS1117 (0.2 μM final concentration). **(B)** Location of mutant sites in the CBS1117-H5 HA structure. Mutants exhibiting >60% entry in the presence of inhibitor (i.e., those that are relatively resistant to CBS1117 inhibition) are colored red and mutants exhibiting <50% entry in the presence of the inhibitor (i.e., those that are still sensitive to CBS1117 inhibition) are colored green. **(C)** Location of escape mutations mapped on the structure of the CBS1117-H5 HA complex. CBS1117 is shown in green and residues observed as escape mutations in H1 HA are shown in yellow.

interactions, including those of HA2-W21 and HA2-I45, which are presumably partially responsible for the >10× decrease in potency (Fig 4).

CBS1117 has greater potency against group 1 HA strains, including circulating H1 HA and avian H5 HA (Gaisina et al, 2020; Hussein et al, 2020). Our laboratories and others have noted that HA fusion inhibitors generally exhibit group specificity. Examples include group 1 fusion inhibitors MBX2329 (EC$_{50}$ ~0.5 μM against H1 HA versus EC$_{50}$ > 100 μM against H3 HA [Basu et al, 2014]), MBX2546 (EC$_{50}$ ~0.3 μM against H1 HA versus EC$_{50}$ > 100 μM against H3 HA [Basu et al, 2014]), and CBS1117 (EC$_{50}$ ~3 μM against H5 HA versus EC$_{50}$ > 50 μM against H3 HA [Gaisina et al, 2020]). In contrast, Arbidol is an interesting exception in that it targets a wide range of enveloped viruses including both group 1 and 2 HA of influenza (i.e., it is a promiscuous binder), albeit at relatively low potency against influenza (EC$_{50}$ > 8 μM [Brancato et al, 2013; Kadam & Wilson, 2017]). In the context of the present work, it is of interest to examine the structural basis for the group specific activity of CBS1117. In Fig 5A, we show the amino acid sequence alignment for the H5 HA residues showing interactions with CBS1117 with H1 HA (group 1) and H3 and H5 HA (group 2). With respect to the residues that show direct interactions with the compound, residues HA1-325, HA2-W21, and HA2-I45 are conserved across the four sequences. Moreover, HA1-Q40 is not conserved and HA2-T49 is weakly conserved between group 1 and 2, suggesting that these residues are not responsible for group specificity. In contrast, HA1-H38 is conserved in group 1 but replaced with a conserved asparagine in H3 and H7 HA. Indeed, HA1-N38 is a highly conserved site for N-glycosylation within group 2 (Wiley & Skehel, 1987). In Fig 5B, we aligned the H3 (group 2) HA structure with our CBS1117-H5 HA structure. As noted above, HA1-H38 interacts directly with CBS1117 via hydrophobic interactions (Fig 2A) and presumably the loss of this interaction at this position results in reduced binding of the compound. Moreover, the presence of a glycosyl group at HA1 position 38 may be expected to further diminish compound binding and potency in group 2 HA due to steric hindrance. Together, the loss of a hydrophobic interaction and

presence of steric hindrance provide a likely mechanism for the observed group specificity. Glycosylation at this site has been recently proposed to be partially responsible for the group specificity of JNJ4796 (van Dongen et al, 2019), a fusion inhibitor that binds to a site that partially overlaps with the CBS1117-binding site.

The H5 HA-CBS1117 structure represents the fourth crystal structure of a small-molecule inhibitor bound to HA. Previous complex structures include the fusion inhibitors TBHQ (Russell et al, 2008) and Arbidol (Kadam & Wilson, 2017), which inhibit group 2 HA and bind close to the HA stem loop. On the other hand, the fusion inhibitor JNJ4796 (van Dongen et al, 2019) binds to a region that overlaps with the CBS1117-binding site near the fusion peptide. For example, the piperidine/piperazine and chlorobenzyl/benzyl rings of the two compounds (Fig 6A) show significant overlap in the structure alignment of the complexes (Fig 6B), with similar protein interactions and modes of actions. However, in the case of CBS1117, the linker between the piperidine and benzyl rings and the

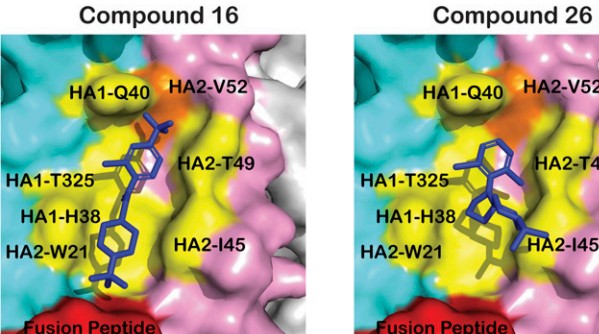

**Figure 4. Structure–activity relationship of derivatives of CBS1117.**
The activity of the compounds has been previously described (Gaisina et al, 2020). To generate this figure, the compounds 16 and 26 were manually aligned with CBS1117. The coloring scheme is the same as Fig 2B with the addition of the new potential interaction with the side chain of HA2-V52 in orange.

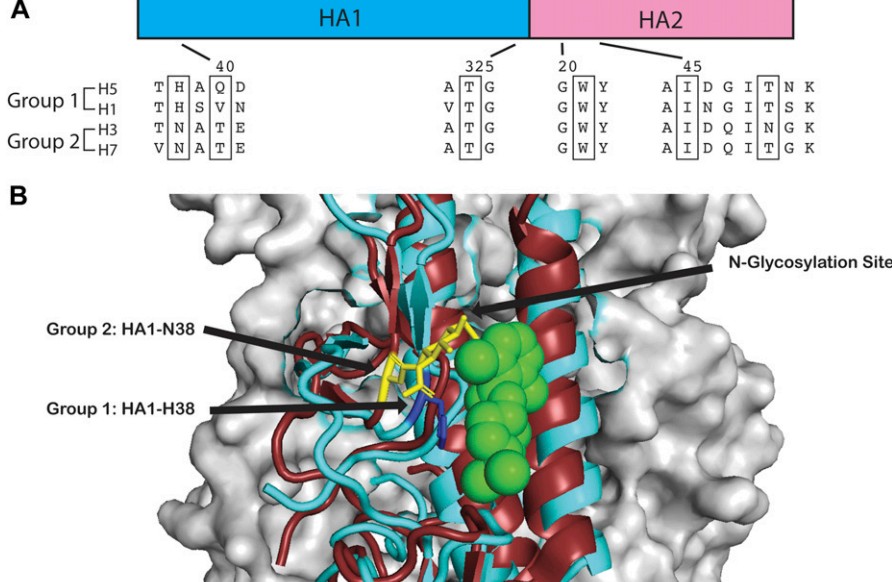

**Figure 5. Group-specific binding of the CBS1117.**
**(A)** Sequence alignment between group 1 (H1 and H5) and group 2 (H3 and H7) HA (35). **(B)** Structure alignment between H5 HA (cyan) in complex with CBS1117 (green) and H3 HA (ruby, PDB entry: 4O5N). Residue 38 is shown in blue for H5 HA and yellow for H3 HA. The N-acetyl-glucosamine observed in group 2 structures is shown in yellow.

interactions between the chlorine of the phenyl moiety with HA2-T49 are clearly unique. Both compounds exhibit similar potencies against infectious group 1 influenza (van Dongen et al, 2019; Gaisina et al, 2020), despite significant differences in their scaffolds. Together, the binding sites for CBS1117 and JNJ4796, as well as the observation that broadly neutralizing antibodies CR6261, FI6v3, and CR9114 bind to this site (van Dongen et al, 2019), underscore the potential for the development of small-molecule therapeutics that target the fusion peptide proximal region.

## Materials and Methods

### Preparation of H5 HA

H5 HA was prepared as previously described (Antanasijevic et al, 2020). Briefly, the H5 HA extracellular domain was expressed in SF9 insect cells grown in SF-900 II serum-free media (Life

Technologies). The cells were co-transfected with a pAcGP67 plasmid containing an H5 HA (A/Vietnam/1203/04 (H5N1)) expression construct and BD BaculoGold linearized baculovirus DNA (BD Biosciences). In the H5 HA construct, the mammalian cell secretion signal sequence (residues M1 to S16) was replaced with the GP67 secretion signal (pAcGP67 vector). Transmembrane and cytosolic regions of HA (residues V521 to R564 of HA0) were removed and replaced with an artificial trimerization domain (the foldon from T4 fibritin) and a His-tag. Cell handling, transfection, and protein expression were performed as recommended by the BD BaculoGold starter package kit (BD Biosciences). Viral titers were monitored using the BacPAK qPCR Titration Kit (Clontech Laboratories). For expression, fresh SF9 cells at $3-4 \times 10^6$ cells/ml confluency were infected with H5 HA–containing baculovirus at MOI between two and four. 4 d later, the suspension was collected, and the cells were removed by centrifugation. H5 HA is secreted into the insect cell media and purified by Ni-affinity chromatography. Given the relatively low processing percentage of HA expressed in this

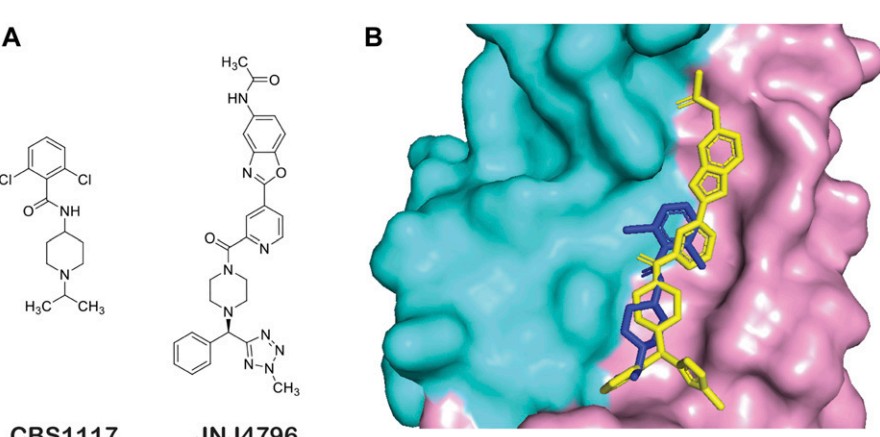

**Figure 6. Comparison of group 1 fusion inhibitors CBS1117 and JNJ4796 bound to H5 HA.**
**(A)** Comparison of chemical structures of CBS1117 and JNJ4796. **(B)** Alignment of bound complexes with CBS1117 colored green and JNJ4796 (PDB entry: 6CFG) colored yellow. In the surface representation of H5 HA, the HA1 and HA2 subunits are colored cyan and pink, respectively.

insect cell line, furin protease (NEB) was used to cleave the HA0 into HA1 and HA2. After 48 h at 4°C, furin was inactivated using furin inhibitor I from EMD Millipore. The protein concentrate was then subjected to Sephacryl S300 gel filtration column with phosphate buffer (50 mM NaPO$_4$ [pH 8.1] and 50 mM NaCl) as a running buffer. Protein fractions were pooled and concentrated, and the final yield was ~3 mg of protein per liter of SF9 cells.

### NMR experiments

WaterLOGSY and STD experiments were performed, as previously described (Antanasijevic et al, 2014a), on a Bruker 800 MHz AVANCE spectrometer equipped with a room temperature triple resonance probe. In the case of WaterLOGSY, the relaxation delay was 2.5 s, the mixing time was 2 s, and the number of scans was 1,024 (experimental time ~80 min). Experimental conditions were 100 $\mu$M CBS1117 ± 5 $\mu$M H5 HA in 20 mM NaPO$_4$ (pH 7.4), 150 mM NaCl, and 10% $^2$H$_2$O at 25°C. In the case of STD, the relaxation delay was 2.5 s, the mixing time was 1 s, and the number of scans was 4,096 (on-resonance at −1 ppm and off-resonance at 56 ppm, experimental time ~8 h). Experimental conditions were 100 $\mu$M CBS1117 + 10 $\mu$M H5 HA in 20 mM NaPO$_4$ (pH 7.4), 150 mM NaCl, and 100% $^2$H$_2$O at 25°C. Data were processed and analyzed using NMRPipe and NMRDraw (Delaglio et al, 1995).

### X-ray crystallography

For the crystallography experiments, the C-terminal foldon domain (and His-tag) of HA2 was removed by the addition of thrombin at 10–50 ng/ml, followed by purification by size exclusion chromatography (Sephacryl S300 gel filtration column using 50 mM NaPO$_4$ [pH 8.2] + 50 mM NaCl as the running buffer). HA was then concentrated to 8 mg/ml and subjected to multiple crystallization tests. The structure of H5 HA in complex with CBS1117 was collected from crystals grown in reservoir containing 100 mM Tris (pH 8.5) + 20% PEG 6000 + 10% Glycerol. The drops were initially set by mixing 1 $\mu$l of 8 mg/ml H5 HA solution and 1 $\mu$l of reservoir solution (using the hanging drop method). Crystals were soaked in a cryosolution containing 100 mM Tris (pH 8.5) + 20% PEG 6000 + 20% Glycerol + 5 mM CBS1117. Diffraction datasets were collected at the stations of Life Sciences Collaborative Access Team at the Advanced Photon Source, Argonne, Illinois. Initial data processing was performed using XDS. Molecular replacement and structure refinement were performed in CCP4 (Winn et al, 2011) using PDB entry:2FKO as the starting model. Automatic refinement was performed with REFMAC5, with manual refinement carried out using Coot (Emsley et al, 2010). J-Ligand was applied to generate structure files and restraints for CBS1117, and the ligand was then introduced in the complex structure using Coot. The PyMOL program package was used for structure comparison and generation of figures. Ligplot$^+$ (Laskowski & Swindells, 2011) was used to determine hydrophilic and hydrophobic interactions between CBS1117 and HA crystals.

### Viral entry assays

Mutagenesis and the viral entry assays were performed as previously described (Antanasijevic et al, 2014b). Briefly, A549 lung cells, which were maintained in Dulbecco's medium with 10% FBS and 1%

penicillin–streptomycin, were seeded to 2 × 10$^4$ cells/well of a 24-well cell culture plate in a volume of 0.5 ml. The following day, 500 $\mu$l of the virus stock, in the presence or absence of CBS1117 at 2× EC$_{50}$ (0.2 $\mu$M final concentration), was added to each of the wells of the A549 cells after removal of the medium. The plates were incubated in a 10% CO$_2$ incubator at 37°C. After ~6 h, the virions were aspirated and replaced with fresh medium, and the cells were allowed to rest for another 48 h. Luciferase activity, which fell within the linear range of detection (i.e., the values of the wild-type and mutants never exceeded 3 × 10$^6$ relative light units), was measured using the Luciferase Assay System from Promega and a Berthold FB12 luminometer running Sirius software.

### Generation of resistance mutants

Resistant viruses (A/PR8/34(H1N1)) were selected by in vitro serial passages in A549 cells in the presence of increasing concentrations of CBS1117, starting with the IC$_{50}$ concentration (0.1 $\mu$M). Viral titers after each passage was assessed by plaque assay on MDCK cells. The virus was harvested 72 hpi, and supernatant was used to infect a new A549 cells. Every two passages the concentration of the drug was increased by fivefold. This was repeated until the drug concentration reached 20 $\mu$M. The presence of drug-resistant virus population was confirmed by performing fitness experiment in A549 cells (MOI-0.01) with or without the drug (20 $\mu$M). In parallel, control passaging in A549 in the presence of DMSO. After nine passages, viral RNA was isolated from the supernatants using a QIAmp viral RNA mini kit (QIAGEN). A SuperScript III one-step RT-PCR kit (Invitrogen) was used to amplify HA segment using 5′-AGCAAAAGCAGGGGAAAA-TAAAAACAACC-3′ and 5′-AGTAGAAACAAGGGTGTTTTTCCTCATATC-3′ primers. RT-PCR product was subsequently cloned into pGEM-T vector and 20 individual clones were sequenced at University of Chicago DNA sequencing facility.

### Accession numbers

Coordinates and structure factors have been deposited in the Protein Data Bank with accession number 6VMZ.

# Supplementary Information

# Acknowledgements

This research was supported by the Chicago Biomedical Consortium with support from the Searle Funds at the Chicago Community Trust to M Caffrey and A Lavie, National Institutes of Health (NIH) R41AI127031 and R41AI145727 to L Rong, and NIH R01AI123359 and R01AI127775 to B Manicassamy.

### Author Contributions

A Antanasijevic: formal analysis, investigation, methodology, and writing—original draft.

MA Durst: formal analysis and investigation.
H Cheng: formal analysis and investigation.
IN Gaisina: formal analysis and investigation.
JT Perez: formal analysis and investigation.
B Manicassamy: formal analysis and methodology.
L Rong: formal analysis and writing—original draft.
A Lavie: formal analysis, investigation, and writing—original draft.
M Caffrey: conceptualization and writing—original draft, review, and editing.

## Conflict of Interest Statement

The authors declare that they have no conflict of interest.

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
