## [Reviewer comments · Life Science Alliance]

Life Science Alliance

Structure of Avian Influenza Hemagglutinin in Complex with a Small Molecule Entry Inhibitor

Aleksandar Antanasijevic, Matthew Durst, Han Cheng, Irina Gaisina, Jasmine Perez, Balaji Manicassamy, Lijun Rong, Arnon Lavie, and Michael Caffrey

DOI: <https://doi.org/10.26508/lsa.202000724>

Corresponding author(s): Michael Caffrey, University of Illinois at Chicago

Review Timeline:

Submission Date:	2020-03-31
Editorial Decision:	2020-05-04
Revision Received:	2020-06-05
Editorial Decision:	2020-06-09
Revision Received:	2020-06-12
Accepted:	2020-06-15

Transaction Report:

May 4, 2020

Re: Life Science Alliance manuscript #LSA-2020-00724-T

Prof. Michael Caffrey
University of Illinois at Chicago
Biochemistry and Molecular Biology
Chicago 60607

Dear Dr. Caffrey,

Thank you for submitting your manuscript entitled "Structure of Avian Influenza Hemagglutinin in Complex with a Small Molecule Entry Inhibitor" to Life Science Alliance. The manuscript was assessed by expert reviewers, whose comments are appended to this letter.

As you will see, the reviewers think that your work is of value to the field. They further provide constructive input on how to strengthen your manuscript. We would thus like to invite you to submit a revised version, addressing the concerns of the reviewers. Importantly, a major re-writing and re-structuring is needed - please follow reviewer #1's suggestions to do so. The concerns of this reviewer regarding figure 4 need to get addressed, too. Further, if you have data at hand to address the comments of reviewer #2 regarding figure 3 and regarding inhibitor comparisons via NMR, please add these.

The typical timeframe for revisions is three months, but we have updated our policies in light of the current pandemic. Please get in touch in case you need more time. Please note that papers are generally considered through only one revision cycle, so strong support from the referees on the revised version is needed for acceptance.

Thank you for this interesting contribution to Life Science Alliance. We are looking forward to receiving your revised manuscript.

Sincerely,

B. MANUSCRIPT ORGANIZATION AND FORMATTING:

Reviewer #1 (Comments to the Authors (Required)):

Antanasijevic et al report the structure of HA from A/Vietnam/1203/04 (H5N1) bound with a small-molecule entry inhibitor, CBS1117. The structure accounts for previously determined (and

published) SAR and for resistance mutations selected for as described in this MS. As this is just the fourth report of HA with bound entry inhibitor, it deserves publication in a journal like *J. Virol*, ordinarily scanned by the virology community and those interested in topics such as influenza virus. The structure itself appears to be well determined, at least as deduced from the statistics in Table 1.

The paper would benefit greatly from condensation into a relatively short note, as follows:

- (1) Shorten the introduction substantially. It reads like a graduate-school primer on HA fusion. Anyone who reads this paper will know about all that (or should). Moreover, the opening sentence and much of what is in the first paragraph are known to every reader of the *NY Times*. The correct Introduction would not bore the reader with statistics about world-wide case fatality rates. Instead, it should start by pointing out that there are only three targets of currently used influenza-virus inhibitors (M2, NA, and endonuclease), indicate that each can select for resistance and use of the M2 and NA inhibitors has already led to widespread resistance. It might point out that in other cases (e.g., HIV) monotherapy almost always elicits resistance (in the case of HIV, within an individual, while in the case of flu or familiar antibiotics, in the population). That observation can then motivate study of additional targets -- in this case, HA. Hence CBS1117. All that can be said in two carefully written paragraphs.
- (2) Results. Move the NMR section to some sort of supplement. It reads like a throwback to the 20th century, when NMR spectroscopists kept wanting to claim that something seen in a crystal needed to be confirmed "in solution". I believe that the world has largely gotten over that one. The crystal structure and the mutations bridge from chemistry to biology appropriately, and the NMR data are at best part of a self-consistency argument. Don't waste the reader's time. If the NMR experiments could have shown some potential cooperativity among the three sites or some other physico-chemical characteristic of binding, my recommendation might have been different.
- (3) Discussion. Some modest, less drastic condensations are probably possible.

One crucial cluster of experimental points that need to be addressed before publication anywhere. Figure 4: The effects of the mutations in fig. 4a do not seem particularly strong, and the caption does not give the concentration of CBS1117. Why was the experiment not done as a function of drug concentration? Indeed, what does the binding isotherm of the compound look like (perhaps it is in an earlier paper -- if so, make a reference to it and give the K_d)? Likewise, where are the quantitative data on resistance by the selected mutations?

One more general question: has CBS1117 been shown directly (with a virus-liposome fusion assay, for example) to inhibit fusion rather than some other step in entry? Presumably it has, and the binding with HA leaves mechanism in little doubt, but an explicit reference would help.

When rewriting, the authors should pay attention to the following. Many of the points are efforts to cure them of very lazy writing.

- (1) Change "in close proximity to" to "close to" (countless instances) -- they mean the same thing, and the former is just adds Latinic pretension to everyday clarity
- (2) Delete ALL use of "notably", "interestingly", etc. Either the observation is indeed notable, and speaks for itself, or you're trying to doctor up something that isn't really so notable after all. The same is true of "interestingly" and "remarkably": most of the time, my reaction is: "why is that interesting -- seems to me it's obvious" or "why is that remarkable -- on careful analysis, it's quite dull". So all those adverbial intensifiers do to a cautious reader is signal that the writer can't explain the results well enough to make them interesting or notable or remarkable and therefore induce concern that the writer doesn't really know how to think about his/her data.
- (3) Last sentence of next to last paragraph of Intro: prefusion HA is metastable, with respect to the

postfusion conformation, ONLY after cleavage between HA1 and HA2. Make this clear. HA0 is the minimum free energy conformation as the protein folds in the ER, because the postfusion conformation is not possible when the relevant peptide bond is still there. Cleavage changes the accessible configuration space. Misunderstanding on this point vis-a-vis HIV Env has led to lots of strange nonsense in the HIV field (which is beset with strange nonsense of all kinds anyway).

(4) Last paragraph of Intro: change "our groups" to "our laboratories" (anything else), because the double use of "group" (group 1 and 2 subtypes; research groups) reads oddly. Also, in the next to last sentence, "efficiently inhibited replication of infectious H1N1 virus".

(5) Several examples of unnecessary words in the first paragraph of Results:

Delete "As shown by Figure 1b" and simply add (Figure 1b) at the end of the sentence.

Delete "Further analysis reveal that": completely unnecessary

Delete "the formation of" because "by bridging interactions" says the same thing -- and indeed, better, because it is the interactions, not their formation, that matters

Change "in agreement with our group's initial SAR analysis" to "agrees with initial SAR analysis"

(the reference makes it clear who did it: self-reflexive comments just put off discriminating readers - or should)

(6) End of Results: These mutations are very close to the binding site (Figure 4c, residue 323 of HA1 in Leu in H5 HA). In particular, the HA1-T325I mutations eliminates the polar interaction with CBS1117 described above. In summary, the site-directed mutagenesis and resistance experiments are fully consistent with protein-drug interactions at the CBS1117 binding site in H5 HA seen in the x-ray structure and with a similar binding mode in H1 HA.

(7) Bottom of page 10: Site-directed mutagenesis of H5 confirms that drug sensitivity depends on residues close to CBS1117. Moreover, the resistance mutations in H1 HA (another group 1 subtype) selected by propagating virus in the presence of CBS1117 were at four positions immediately adjacent to the CBS1117 binding site in H5 HA. Thus, the drug appears to bind similarly to both HAs of both subtypes.

(8) Bottom of p. 111: Delete sentence starting "As noted above", and (cf. point 2, above) delete "Interestingly" (a cross-group inhibitor would be much more interesting -- and at many sites, surprising).

Reviewer #2 (Comments to the Authors (Required)):

The manuscript by Aleksandar Antanasijevic describes several structural, molecular approaches to understand the determinants of inhibition by CBS1117. The study makes use of previously described inhibitors and mutants from their own work. And eventually far in the discussion, they compared their inhibitor with JNJ4796.

The study is very well performed, and the manuscript describes the experiments efficiently. However, the introduction relies on standard paragraphs, it would be fantastic to see some inhibitor comparisons in the introduction, the good the bad, etc. In BC (before corona) time I would have loved to see the NMR analyses of these different inhibitors.

Major concern,

While we are AC, the omission of the T325x or the introduction of a N-glycosylation site at position 38, in figure 3 would really have finished this story, but now I do not deem them necessary. Especially because T325I pops up during forced viral evolution to generate escape mutants.

Minor

-18 distinct subtypes of HA (H1-H18) and 11 different subtypes of NA (N1-N11) (Nobusawa et al., 1991; Webster et al., 1992)

Wrong references, H16-H18 where not described pre-2000

Dear Editor and Reviewers,

Thank you for your careful reading of our manuscript and the constructive comments. We have revised the manuscript according to reviewers' comments. Enclosed please find the revised manuscript (as well as a markup copy with edits noted), responses to reviewers, and a list of changes made in the manuscript. In addition, we have made a number of changes to fit the journal's format. We sincerely hope that our article will now be acceptable for publication in Life Science Alliance.

Please feel free to contact me at any time.

Sincerely, Michael Caffrey

Reviewer #1:

1. The introduction has been shortened to 2 paragraphs (and the references updated accordingly) as suggested.
2. The NMR results have been moved to the Supplementary Material (new Figure S2) as suggested.
3. (i) The reviewer also questions the relatively small effects observed for the site-directed mutants shown in Figure 4a (now Figure 3ab). The point that we are trying to make is that the mutations that exhibit ~wild-type inhibition (HA2-Q42A and HA2-N53A, shown as green in Figure 4b/now Figure 3b) are apparently not disrupting binding of the compound. On the other hand, the mutants that show large effects, i.e. no inhibition at EC50 (HA2-I45A, HA2-T49A and HA2-V52A, shown as red in Figure 4b/now Figure 3b) are apparently disrupting binding.

(ii) The concentration of the compound (~EC50 in this assay) was listed in the methods, we have added it to the figure caption. In addition, we have cited the original paper for the dose dependent inhibition curve (Hussein et al., 2020).

(iii) The EC50 of the resistance mutants has not been determined. The EC50 of the resistance mutants would be interesting for future studies of HA mutant structures; however, we feel that this is beyond the scope of the present work.

(iv) We have added a reference (Hussein et al., 2020), which demonstrated that this series of compounds are indeed fusion inhibitors, and noted this in the second paragraph of the introduction.

All of the suggestions for stylistic changes were followed:

- (1) "in close proximity to" changed to "close to" (2 occasions).

- (2) "Notably" (3 occasions) and "Interestingly" (5 occasions) have been removed.
- (3) The sentence containing "metastable" was removed to shorten introduction.
- (4) "our groups" changed to "our laboratories" (5 occasions).
- (5) Miscellaneous suggestions for deletions of unnecessary words all followed.
- (6) The last sentences of the Results were re-written as suggested.
- (7) The sentences were re-written as suggested.
- (8) The suggested deletions were made.

Reviewer #2:

Major Concern: As the reviewer notes, further mutational studies would be very interesting, particularly mutants to the glycosylation site at HA1-N38 in group 2, which may be expected to lead to drug sensitivity. We hope to perform these experiments in the future but given the current restraints on our laboratories we are not able to do this at present.

Minor Concern: This sentence was removed from the introduction as suggested by Reviewer #1.

June 9, 2020

RE: Life Science Alliance Manuscript #LSA-2020-00724-TR

Prof. Michael Caffrey
University of Illinois at Chicago
Biochemistry and Molecular Biology
900 S ASHLAND Ave
Chicago, IL 60607

Dear Dr. Caffrey,

Thank you for submitting your revised manuscript entitled "Structure of Avian Influenza Hemagglutinin in Complex with a Small Molecule Entry Inhibitor". We would be happy to publish your paper in Life Science Alliance pending final revisions necessary to meet our formatting guidelines.

- please add your ORCID ID-you should have received instructions on how to do so
- please add a conflict of interest statement to the main manuscript text
- please list 10 authors et al. in the references
- please add callouts to the individual panels for Figure 6

A. FINAL FILES:

B. MANUSCRIPT ORGANIZATION AND FORMATTING:

Sincerely,

Reilly Lorenz
Editorial Office Life Science Alliance
Meyerhofstr. 1
69117 Heidelberg, Germany
t +49 6221 8891 414
e contact@life-science-alliance.org
www.life-science-alliance.org

June 15, 2020

RE: Life Science Alliance Manuscript #LSA-2020-00724-TRR

Prof. Michael Caffrey
University of Illinois at Chicago
Biochemistry and Molecular Biology
900 S ASHLAND Ave
Chicago, IL 60607

Dear Dr. Caffrey,

Thank you for submitting your Research Article entitled "Structure of Avian Influenza Hemagglutinin in Complex with a Small Molecule Entry Inhibitor". It is a pleasure to let you know that your manuscript is now accepted for publication in Life Science Alliance. Congratulations on this interesting work.

DISTRIBUTION OF MATERIALS:

Again, congratulations on a very nice paper. I hope you found the review process to be constructive and are pleased with how the manuscript was handled editorially. We look forward to future exciting submissions from your lab.

Sincerely,

Reilly Lorenz
Editorial Office Life Science Alliance
Meyerhofstr. 1
69117 Heidelberg, Germany
t +49 6221 8891 414
e contact@life-science-alliance.org
www.life-science-alliance.org